# Brain Metastases as Inaugural Sign of Non-Small Cell Lung Carcinoma: Case Series and Review of Literature

**DOI:** 10.3390/cancers16173105

**Published:** 2024-09-08

**Authors:** Alexandra Pușcașu, Fabien Moinard-Butot, Simon Nannini, Cathie Fischbach, Roland Schott, Laura Bender

**Affiliations:** Oncology Department, Institut de Cancérologie Strasbourg Europe, 17 rue Albert Calmette, 67200 Strasbourg, France; f.moinard-butot@icans.eu (F.M.-B.); s.nannini@icans.eu (S.N.); c.fischbach@icans.eu (C.F.); r.schott@icans.eu (R.S.); l.bender@icans.eu (L.B.)

**Keywords:** brain metastases, NSCLC, immune check point inhibitors, oligometastatic

## Abstract

**Simple Summary:**

Treating non-small cell lung cancer (NSCLC) patients with brain metastases (BM) is challenging, especially when brain involvement is the first sign of cancer. This study retrospectively analyzed 25 patients with newly diagnosed brain metastatic NSCLC without EGFR or ALK alterations. The findings suggest that patients with symptomatic BM at diagnosis may have better survival outcomes due to increased use of multimodal local treatments. Combining local approaches with first-line immune checkpoint inhibitors (ICI) and chemotherapy appears to improve survival in these patients. Additionally, a nonsystematic literature review was conducted to better understand the topic and explore the potential benefits of various immunotherapy-based combinations for brain metastatic NSCLC. This research aims to highlight the survival outcomes of this underrepresented population and provide insights into optimal treatment strategies.

**Abstract:**

In the era of immune checkpoint inhibitors (ICI), managing non-oncogene driven non-small cell lung cancer (NSCLC) with brain metastases (BM) is challenging, especially when brain involvement is the initial sign. Patients with newly diagnosed brain metastatic NSCLC without epidermal growth factor receptor (EFGR) nor anaplastic lymphoma kinase (ALK) alterations were retrospectively included. Twenty-five patients were analyzed; 15 (60%) had symptomatic BM as the first sign (group 1), while 10 (40%) had BM discovered during complementary examinations (group 2). Fourteen patients (56%) had concomitant extracerebral metastases, primarily in group 2. Eight (32%) had oligometastatic disease, with seven in group 1. Over half received chemotherapy and pembrolizumab as first-line treatment. BM surgical resection occurred in twelve (80%) patients in group 1 and one in group 2. Median cerebral progression-free survival was 10 months: 12 in group 1 and 5 in group 2. Median overall survival was 25 months: not reached in group 1 and 6 months in group 2. This case series highlights survival outcomes for patients with inaugural BM, a demographic underrepresented in pivotal trials. Oligometastatic disease and symptomatic BM as initial signs seem associated with better prognosis due to increased use of multimodal local approaches. Combining local approaches with first-line ICI+/− chemotherapy appears to improve survival in brain metastatic NSCLC. A literature review was conducted to explore key questions regarding upfront ICI alone or in combination with systemic drugs or local approaches in brain metastatic NSCLC.

## 1. Introduction

Approximately 25% of patients with non-small-cell lung cancer (NSCLC) present brain metastases (BM) at diagnosis [1]. The intracranial efficacy and survival benefit of upfront targeted therapies have been explored in oncogen-driven NSCLC with BM [2,3]. However, data are still scarce regarding the optimal initial approach for BM in NSCLC without targetable alterations mainly in the era of a combination of immune checkpoint inhibitors and chemotherapy as the new standard in metastatic settings. This particular population remains yet to be characterized in terms of optimal therapeutical management, ideal sequence of treatments in the front line (local vs. systemic), and identification of relevant clinical, biological, and molecular factors with an impact on survival.

Currently, the cornerstone of treatment for BM, irrespective of the primary tumor, comprises local approaches, such as surgery and/or radiotherapy. However, these procedures are associated with a long-term negative impact on quality of life due to neurocognitive alterations [4,5], which are non-negligible factors, especially in the era of immunotherapy long responders. Whether using upfront immune checkpoint inhibitors (ICIs) combined with chemotherapy is the key to changing the current paradigm is so far a question to be answered. The pivotal clinical trials that granted the ICI alone or in combination with platinum-based chemotherapy as first-line treatment in NSCLC without targetable mutations allowed the inclusion of a few patients with pretreated, stable, and asymptomatic BM [6,7,8,9]. In the era of standard first-line ICI combined with chemotherapy, and so that some data are in favor of blood–brain barrier (BBM) penetration and intracranial efficacy of ICIs [8,10], it is not clear how to position local approaches in the management of CNS disease.

To better characterize and evaluate the survival outcomes of this specific population in a real-life setting, we conducted a retrospective case series analysis of patients treated for a non-oncogene-driven NSCLC with inaugural BM. Additionally, we conducted a comprehensive literature review focusing on upfront therapeutic management to obtain efficacy data to gain a better understanding of the topic.

## 2. Patients and Methods

Newly diagnosed patients with NSCLC and symptomatic or asymptomatic BM without EGFR nor ALK mutations treated with a first-line systemic therapy (platinum-based chemotherapy or ICIs or combination chemotherapy and ICI) between February 2018 and January 2023, at the Cancer Institute in Strasbourg, France, were included. Patients with leptomeningeal metastases were excluded. The cases were categorized into two distinct groups based on the presence or absence of symptomatic brain involvement as an inaugural sign of the disease. Group 1 included patients with symptomatic BM as the first sign of lung cancer while group 2 comprised patients who had BM discovered during complementary exams within one month of the diagnosis of NSCLC. Brain metastases were confirmed by brain CT scan and/or magnetic resonance imaging (MRI).

Patient data (age, sex, smoking status, performance status), disease characteristics (histology, PD-L1 status, localization, number and dimensions of cerebral lesions, localization of extracerebral lesions), as well as treatment modalities of brain metastases (surgery/radiotherapy) and systemic treatment, were retrospectively extracted from the medical records. Oligometastatic disease was defined as the presence of a maximum of five secondary lesions (cerebral and extracerebral) [11].

Data were analyzed using SPSS software application version 26. Survival data were calculated using Kaplan Meyer analysis. Progression-free survival (PFS) was defined from NSCLC diagnosis to first cerebral or extracerebral progression or death. Cerebral progression-free survival (cPFS) was calculated from NSCLC diagnosis to first cerebral progression or death. Visceral PFS (vPFS) was analyzed from NSCLC diagnosis to first extracerebral lesions or death. Overall survival was defined from NSCLC diagnosis to death of any causes.

A non-systematic literature review was also conducted to gather relevant literature information pertaining to the best management of newly diagnosed brain metastatic NSCLC patients. Searches were performed in databases like PubMed, ScienceDirect, and Google Scholar using keywords such as “NSCLC with brain metastases”, “first-line immune checkpoint inhibitors in NSCLC”, and “management of newly diagnosed NSCLC patient with brain metastases”.

## 3. Results

### 3.1. Patient Population

Twenty-five patients diagnosed with brain metastatic NSCLC between February 2018 and January 2023, without EGFR nor ALK alterations were included. Of those, 15 (60%) had BM as the first symptom of the lung disease (group 1) while 10 (40%) had asymptomatic BM discovered during complementary workup (group 2). Table 1 summarizes patient characteristics. A male predominance was observed, with a median age at diagnosis of 61 years. Group 1 exclusively comprised adenocarcinoma histology, while group 2 included squamous (n = 3) and undifferentiated tumors (n = 1). KRAS mutation was found in 10 patients (40%), with 7 in group 1 and 3 in group 2. Additional mutations included JAK1 (n = 1) and STK11 (n = 1) in group 1, and PI3KA (n = 2) in group 2. Twelve patients exhibited PD-L1 expression exceeding 50%, with 7 from group 1. Fifteen patients (60%) received first-line treatment comprising pembrolizumab (anti-PD-L1) and platinum-based combination therapy. The majority of patients exhibited supratentorial BM localization exclusively (n = 12). Over half of the patients (n = 12) presented metastases ranging from 1 to 5 cm in size while five patients had metastases exceeding 5 cm. In the symptomatic population, local therapies including surgery and radiotherapy were prioritized. Twelve patients (80%) in group 1 underwent surgical intervention, while only one patient in group 2 underwent surgery. Twenty-two (88%) patients received radiotherapy either as a standalone treatment or as a complementary therapy to surgery; 14 (93%) in group 1 and 8 (80%) in group 2. Fifty-six percent of the patients (n = 14) had extracerebral lesions at diagnosis with a predominance of visceral disease in group 2 (n = 6; 40% in group 1 vs. n = 8, 80% in group 2). The group 1 exhibited a predominant prevalence of oligometastatic disease (n = 7; 47%). Post-progression treatments in Group 1 included subsequent radiotherapy (WBRT, n = 2; SBRT, n = 2), systemic therapy with single-agent chemotherapy (n = 5), either alone or in combination with bevacizumab (n = 2), platinum-based combination chemotherapy (n = 1), and the KRAS inhibitor sotorasib (n = 2).

### 3.2. Survival Data

The median follow-up time was 22 months. More than half of the patients experienced cerebral progression in both groups; eight (53%) in group 1 and six (60%) in group 2. Visceral progression occurred in eight (54%) patients in group 1 and seven (70%) in group 2. In the study, five patients from group 1 were still receiving first-line treatment, while all patients from group 2 had progressed.

Median PFS for all cases was 6 months. The patients in group 1 exhibited a median PFS of 9 months compared to 3 months in group 2. The median cPFS for the entire population was 9 months; with a median of 12 and 5 months in groups 1 and 2, respectively. The median vPFS was 20 and 7 months in groups 1 and 2, respectively.

At the time of analysis, half of the overall population had deceased (n = 13; 52%), with 8 (53%) from group 2 and 5 (50%) from group 1 (*p* = 0.02). The median OS in all cases reached 25 months. The median OS was not reached in group 1; while the patients in group 2 had a 6-month median OS. As oligometastatic disease was predominant in group 1, this feature could explain the better survival outcomes. Survival data are summarized in Table 2.

## 4. Discussion

Brain metastases in newly diagnosed non-oncogene-driven NSCLC patients are associated with an unfavorable prognosis [1]. Current treatments mainly involve local approaches (surgery/radiotherapy) due to drug penetration limitations through the blood-brain barrier. First-line immunotherapy, alone or combined with chemotherapy, shows promise for non-oncogene-driven NSCLC with brain metastases. The exact mechanism of immunotherapy’s effectiveness in the brain remains unclear, possibly involving peripheral immune cell activation [12].

This case series highlights that oligometastatic disease and symptomatic brain as first signs of lung cancer are associated with a better prognosis. These results seemed to be explained by increased use of brain local treatments based on surgery and/or stereotaxic radiotherapy in combination with standard first-line systemic ICI+/− chemotherapy.

The observed median PFS of 6 months was similar to literature data which reported a median PFS between 2.3 and 12.5 months in studies involving cohorts of non-mutated NSCLC patients with brain metastases treated with ICI alone or in combination with chemotherapy as first-line therapy (Table 3 and Table 4). The median cerebral PFS of 10 months in our series exceeded that reported in other studies [13,14]. Overall survival reached 25 months, two other retrospective studies, including 176 and 23 patients with symptomatic or asymptomatic brain metastatic NSCLC with a PDL1 expression over 50% treated by pembrolizumab, reporting an OS superior to 20 months (29.5 months and 21.6 months) [2,15]. In our cohort, these favorable survival data could be explained because 32% of the patients had an oligometastatic disease. This particular population had a better prognosis thanks to brain surgery and/or stereotaxic radiotherapy that improves brain local control. Frost et al. reported the same result in a retrospective study including 103 patients with exclusive oligometastatic brain metastases of lung cancer. These patients experienced brain long-term disease control and subsequently favorable survival thanks to repeat local ablative therapy [16].

Moreover, while the small sample size of cases precludes direct comparison between patients with symptomatic and asymptomatic brain metastases at diagnosis, certain trends may be observed. Patients with inaugural symptomatic brain metastases appear to demonstrate better PFS and OS outcomes compared to those with asymptomatic brain metastases at NSCLC diagnosis. This observation could be attributed to the prompt implementation of local treatment modalities owing to the presence of symptomatic cerebral disease, which likely contributed to delayed disease progression or relapse in secondary brain lesions. Since most of these patients underwent brain radiotherapy (stereotactic radiosurgery or whole-brain radiotherapy), the possibility of an abscopal effect providing a rationale for these results is plausible [31]. In addition, as suggested by previous studies, radiotherapy may facilitate blood-brain barrier penetration, thereby enhancing the delivery of treatment molecules [32]. Several literature data suggested that a combination of local approaches and immunotherapy is associated with an improvement of local control and overall survival in brain metastatic lung cancer populations [32,33]. Another important question to consider is whether the difference in survival between the two groups could be attributed, at least in part, to the different treatments patients received after progression, such as subsequent radiotherapy or targeted therapy. A more extensive patient population is necessary to thoroughly explore this hypothesis.

To conclude, while the limited sample size necessitates cautious interpretation and precludes generalization to broader populations, several notable findings emerged from this case series analysis. Oligometastatic disease and inaugural symptomatic brain metastases are associated with a better prognosis thanks to increased use of local approaches including surgery and radiotherapy that favorably impact overall survival. Brain local approaches combined with ICI+/− chemotherapy seemed to be the optimal therapeutic management in the brain metastatic lung cancer population. Further research with larger sample sizes is needed to validate these observations.

A comprehensive review of current available data on ICI mechanism of action against brain metastases, brain efficacy data of first-line ICI alone, ICI in combination with chemotherapy or other drugs, and ICI in combination with local approach mainly radiotherapy in non-oncogene driven NSCLC with BM is depicted below.

## 5. State of Art in Newly Diagnosed BM NSCLC

### 5.1. ICI’s Mechanism of Action in BM

Immunotherapy’s efficacy in NSCLC BM is not fully understood, with debates on whether it directly crosses the blood–brain barrier (BBB) or indirectly influences peripheral immune cells.

As brain metastases form, the BBB weakens, allowing cancer cells to create an inflammatory environment. This attracts and activates brain cells, promoting tumor growth and BBB disruption [12]. Literature indicates limited antibody uptake in the CNS under normal conditions [34]. However, enzyme-linked immunosorbent assay analysis showed measurable concentrations of nivolumab and pembrolizumab in both serum and cerebrospinal fluid (CSF), suggesting possible CNS penetration [10]. Evidence of increased BBB permeability and enhanced vesicular transport in brain metastatic breast cancer has equally been reported [35]. These findings might be suggestive of a higher CNS penetration of molecules, including ICI agents, during BM formation.

When examining rodent models, ICIs’ efficacy at the cerebral level appears to rely more on systemic immune responses [36]. Additionally, murine experiments have shown ICIs’ ability to promote the trafficking of NK cells and CD8 T cells in the CNS, with immune cell infiltration density at BM associated with increased OS [37]. These findings suggest an indirect intracranial immunotherapy effect, driven by local CNS interactions among cancer immune cells.

### 5.2. Assessing the Effectiveness of First-Line Treatment Modalities

#### 5.2.1. First-Line Single or Dual Immunotherapy Agents

In NSCLC, survival benefits of ICIs for newly diagnosed patients with BM are primarily gleaned from subgroup analyses of larger phase III clinical trials. In the Checkmate 227 trial, patients with BM treated with nivolumab and ipilimumab exhibited longer OS compared to the chemotherapy arm (16.8 vs. 13.4 months) [38]. A recent 5-year post-hoc analysis of the same trial demonstrated durable survival outcomes in both BM and non-BM patients following first-line double ICI treatment, with BM patients achieving an OS of 17.4 months (HR 0.63). Favorable trends in systemic (HR = 0.77, 95% CI: 0.51–1.15) and intracranial PFS (HR = 0.82, 95% CI: 0.52–1.30) were observed with nivolumab and ipilimumab combination treatment. Moreover, fewer patients experienced new BM in the double ICI arm compared to the chemotherapy arm (4% vs. 20%) [21].

A recent pooled analysis of four key NSCLC trials, including pembrolizumab in both upfront (KEYNOTE 024 and KEYNOTE 042) and second-line (KEYNOTE 001 and KEYNOTE 010) settings, highlights an overall survival benefit regardless of brain metastasis presence or PD-L1 expression levels. In patients with PD-L1 expression ≥ 1%, the OS benefit was evident with a median OS of 13.4 months compared to 10.3 months with chemotherapy. For those with PD-L1 > 50%, the median OS was 19.7 months vs. 9.7 months with chemotherapy [19].

A subgroup analysis of the EMPOWER-Lung 1 trial, examining cemiplimab, an anti-PD-1 ICI, revealed superior outcomes in first-line monotherapy for NSCLC patients with BM, showing longer median OS, PFS, higher ORR, and longer duration of response compared to chemotherapy (OS not reached vs 20.7 mo, PFS 12.5 vs 5.3 mo, ORR 55.9 vs 11.4% [17,39]. Similarly, in the IPSOS study, atezolizumab demonstrated improved OS regardless of histology, PD-L1 expression level, or ECOG PS, with a doubled 2-year OS rate compared to single chemotherapy (24.3% vs. 12.4%), including benefits in the BM subgroup (HR for OS of 0.85) [18].

Prospective trials mostly enrolled pretreated asymptomatic BM patients. One trial specifically assessed single-agent ICI efficacy and safety in untreated, asymptomatic BM from NSCLC and melanoma, though not in the first-line setting [40]. The final analysis in 42 NSCLC patients showed a BM response rate of around 30% in PD-L1 > 1% positive cases, with median intracranial PFS and OS of 2.3 and 9.9 months, respectively. Some experienced CNS progression despite extra-cerebral lesion response, possibly due to differing PD-L1 expression in the brain [40].

Various retrospective studies, alongside pivotal trials, examined first-line ICI outcomes and potential prognostic factors. A French multicentric study found no significant difference in survival outcomes and overall response rates in PD-L1 > 50% of patients with and without BM treated with upfront pembrolizumab monotherapy [2]. Conversely, a larger retrospective analysis indicated a potential negative impact of BM and squamous histology, with significantly shortened OS regardless of treatment regimen, regardless of treatment regimen [23].

To conclude, ICI as a single or dual agent seems to be an effective therapy for the brain metastatic NSCLC population. Table 3 encompasses first-line setting studies, focusing on patients treated with ICI alone.

#### 5.2.2. First-Line ICI and Chemotherapy

Subgroup analysis of KEYNOTE 189 study [41] (first-line pembrolizumab + platinum-based chemotherapy vs. chemotherapy) and CheckMate 9LA study [8,42] (first-line nivolumab + ipilimumab and chemotherapy vs. chemotherapy) confirmed an improvement in both OS and PFS for the BM population in the ICI arm, compared to the chemotherapy alone arm, regardless of PD-L1 status. In CheckMate 9LA, OS was estimated at 19.3 months in the ICI/chemotherapy group vs. 6.8 months in the chemotherapy group, while systemic PFS reached 9.7 months compared to 4.1 months, respectively [8]. These findings were confirmed by the updated analysis of KEYNOTE-189, with a reported median OS of 19.2 months in the BM group treated with ICI/chemotherapy vs. 7.5 months in the chemotherapy arm and a median systemic PFS of 6.9 in the pembrolizumab/chemotherapy arm vs. 4.7 months in the chemotherapy arm [41]. OS was in favor of ICI/chemotherapy with similar results among patients with (HR 0.41; 95% CI, 0.24 to 0.67) and without (HR 0.59; 95% CI, 0.46 to 0.75) brain metastases.

Moreover, a pooled analysis of phase III trials KEYNOTE 189, KEYNOTE 021 cohort G, and KEYNOTE 407, that enrolled chemotherapy naïve NSCLC, highlighted a prolonged median OS (18.8 months) and PFS (7.6 months) in BM patients in the pembrolizumab plus chemotherapy arm, with higher response objective rates compared to chemotherapy alone [43]. In addition, clinical outcomes were improved with ICI regardless of the presence of BM, and across all PD-L1 groups (<1% included). Both KEYNOTE 189 and KEYNOTE 407 allowed the enrollment of untreated BM patients with lesions < 1.5 cm, as long as they were asymptomatic and without corticotherapy requirement [43].

In the IMPOWER 132 trial, first-line atezolizumab combined with carboplatin or cisplatin plus pemetrexed showed a PFS benefit in patients with metastatic non-squamous NSCLC previously treated. Asymptomatic BM patients were allowed to participate. However, there was no final subgroup analysis for this particular population [44]. A single-arm phase II prospective clinical trial (ATEZO-BRAIN) evaluated the benefit of a chemo-immunotherapy combination for NSCLC patients with untreated BM [24]. Ninety-three percent of patients had synchronous BM at lung cancer diagnosis, and fifty-five percent were receiving corticosteroids at baseline (maximum 4 mg of dexamethasone/day). An intracranial ORR was observed in 40% of patients, including four complete responses. Median intracerebral PFS was 6.9 months with a median OS of 11.8 months. Discordant responses between the CNS and the extra-cerebral disease were equally observed, the underlying supposition being the difference in PD-L1 expression between visceral and cerebral lesions. Table 4 encompasses first-line setting studies, focusing on patients treated with ICI in combination with chemotherapy.

#### 5.2.3. ICIs and Local Approaches: Sequence of Treatments in First-Line

Despite some positive outcomes in small-scale trials [23,26,45], the effectiveness of systemic therapy alone for NSCLC patients with untreated brain metastases remains uncertain. A meta-analysis involving 566 NSCLC patients did not show significant differences in intracerebral response rates between those treated with radiotherapy and immunotherapy vs. immunotherapy alone [46]. Therefore, while immunotherapy alone appears effective, its exclusive use in first-line treatment requires further validation, especially considering that most patients in the analysis received it as a second-line option.

Exploring concurrent radiotherapy and immunotherapy is ongoing, but results vary. One retrospective study of NSCLC patients receiving ICIs within 4 weeks of stereotaxic radiotherapy (SRT) showed no significant survival benefit. However, another analysis found improved overall survival when ICIs were given within 30 days of stereotactic radiosurgery (SRS), especially in PD-L1 positive NSCLC patients (40 months vs. 8 months in the control group) [47]. The same advantage in OS with concomitant SRS and ICIs was seen equally when immunotherapy was administered within 2 weeks of SRT [48]. Yet, these studies lack consensus on treatment timing and comparison arms, leaving the optimal treatment sequence for newly diagnosed patients with brain metastases uncertain.

In the age of prolonged treatment responses, the long-term neurocognitive effects of local brain metastase therapies like surgery or radiotherapy are concerning. While survival rates improve, the impact on quality of life from potential neurological impairments persists. Deciding whether to forego local treatments or consider alternative treatment sequences in the first line remains an unanswered question.

#### 5.2.4. ICIs and Other Agents

Neoangiogenesis, a critical pathway in tumor progression, is primarily regulated by vascular endothelial growth factor (VEGF) signaling [49]. Clinical trials have shown improved survival with VEGF inhibitors like bevacizumab compared to chemotherapy alone in first-line NSCLC treatment (median OS 12.3 vs. 10.3 mo), but these studies did not include patients with brain metastases [3].

In 2018, the IMPOWER 150 study examined the potential synergy between anti-angiogenic drugs and ICIs. It demonstrated improved PFS (8.3 mo) and OS (19.2 mo) with atezolizumab plus bevacizumab plus chemotherapy (ABCP)compared to bevacizumab plus chemotherapy (BCP) in first-line treatment for non-squamous NSCLC (PFS 6.8 mo and OS 14.4 mo, respectively) [50]. In a recent phase III study (ONO-4538-52/TASUKI-52), a first-line treatment regimen of nivolumab plus bevacizumab and chemotherapy demonstrated significantly longer PFS compared to placebo (12.1 months vs. 8 months). This survival benefit was also observed in patients with brain metastases (N = 77), with a PFS of 10.5 months in the nivolumab arm vs. 7.1 months in the placebo arm, irrespective of PD-L1 expression [30].

### 5.3. Future Perspectives

Managing newly diagnosed NSCLC patients with BM poses challenges with many unresolved aspects. While current evidence favors ICIs alone or in combination for better outcomes, notably in pretreated and asymptomatic patients, their efficacy without local treatments remains uncertain. In ATEZO BRAIN, a phase II clinical trial that allowed untreated BM, an intracranial response rate of 42.7% was observed with an OS of 11.8 months [24]. In an era of immunotherapy, avoiding local treatments may spare long-term cognitive effects, yet the optimal treatment sequence remains unclear. Additionally, concurrent radiotherapy and immunotherapy raise concerns about radionecrosis risk.

Furthermore, an inquiry arises regarding the optimal partner for ICIs to enhance intracranial efficacy. For example, VEGF inhibitors showed promising results in first-line settings, BM population included [30,50]. Identifying the patients who could derive significant survival advantages from this therapeutic combination would definitely be a key point in the management of this special population.

In addition, the identification of precise predictive biomarkers for the intracranial efficacy of ICI in NSCLC remains to be clearly defined. Proposed biomarkers include PD-L1 expression, tumor-infiltrating lymphocytes, and tumor mutational burden, but we are still far from being able to select the patients with cerebral disease that would benefit the most from ICIs.

Several ongoing studies are currently trying to approach the unmet needs of NSCLC patients with BM. NIVIPI-Brain is a phase II clinical trial evaluating the double immunotherapy nivolumab/ipilimumab with 2 cycles of platinum-based chemotherapy in newly diagnosed NSCLC patients with untreated BM (NCT05012254 trial). Other trials analyze the combination and timing of local therapies with ICIs (NCT04650490 trial, NCT02696993 trial). Most certainly, continuing to conduct international trials would be the key to finding the optimal management of this special category of patients.

## 6. Conclusions

The occurrence of BM as the inaugural sign of NSCLC represents a unique subset that has not been formally included in pivotal trials evaluating ICI alone or in combination with other systemic therapies. This case series highlights oligometastatic disease and inaugural symptomatic brain metastases that seemed to be associated with a better prognosis thanks to the use of local ablative therapies (surgery/stereotaxic radiotherapy). Brain local approaches combined with ICI+/− chemotherapy seemed to be the optimal therapeutic management in the brain metastatic lung cancer population. Further research with larger sample sizes is needed to validate these observations. In addition, several unresolved questions persist, encompassing considerations regarding the appropriate sequence of treatments in the first line, the identification of potential partners for ICI in BM populations, the determination of patients who would derive the greatest benefit from a specific therapeutic combination, and the identification of predictive biomarkers for ICI efficacy in the brain. Comprehensive data reporting outcomes of treatments in the first-line setting for this population is imperative for informing routine clinical practice.

## Figures and Tables

**Table 1 cancers-16-03105-t001:** Patient characteristics in entire case series and subgroups: group 1: Patients with symptomatic BM at diagnosis; group 2: Patients with BM diagnosed during workup.

	All Cases	Group 1	Group 2
	n = 25	n = 15 (60%)	n = 10 (40%)
Male sex, n (%)	18 (72)	10 (66)	8 (80)
Median Age at diagnosis (years)	63.08 ± 10.17	61.73 ± 10.16	65.10 ± 10.37
Smoking status at diagnosis, n (%)			
Current	8 (32)	4 (26)	4 (40)
Former	17 (68)	11 (73)	6 (60)
Histology, n (%)			
Adenocarcinoma	21 (84)	15 (100)	6 (60)
Squamous carcinoma	3 (12)	0	3 (30)
Undifferentiated	1 (4)	0	1 (10)
ECOG, n (%)			
0–1	22 (88)	15 (100)	7 (70)
≥2	3 (12)		3 (30)
PD-L1 status, n (%)			
<1%	7 (28)	6 (40)	1 (10)
1–49%	6 (24)	2 (13)	4 (40)
≥50%	12 (48)	7 (47)	5 (50)
Molecular profile, n (%)			
KRAS mutation	10 (40)	7 (46)	3 (30)
1st line treatment, n (%)			
Pembrolizumab monotherapy	5 (20)	2 (13)	3 (30)
Pembrolizumab + histology based CHT	15 (60)	9 (60)	6 (60)
Histology based CHT alone	5 (20)	4 (27)	1 (10)
Location of BM at diagnosis, n (%)			
Subtentorial lesions only	7 (28)	5 (33)	2 (20)
Supratentorial lesions only	12 (48)	8 (53)	4 (40)
Both	6 (26)	2 (13)	4 (40)
Number of BM at diagnosis, n (%)			
Single lesion	8 (32)	6 (40)	2 (20)
Between 1 and 5	12 (48)	8 (53)	4 (40)
≥5	5 (20)	1 (6)	4 (40)
Largest diameter of BM n (%)			
<1 cm	3 (12)	0	3 (30)
1–5 cm	19 (76)	13 (87)	6 (60)
>5 cm	3 (12)	2 (13)	1 (10)
Patients with extracerebral metastases, n (%)	14 (56)	6 (40)	8 (80)
Bone	2 (14)	0	2 (20)
Lung	2 (14)	1 (17)	1 (10)
Other localisation	4 (28.5)	2 (33)	2 (20)
Mixt	6 (42.8)	3 (50)	3 (30)
Brain surgery, n (%)	13 (52)	12 (80)	1 (10)
Complete	8 (38)	8 (67)	0
Incomplete	5 (61)	4 (33)	1 (100)
Brain RT, n (%)	22 (88)	14 (93)	8 (80)
WBRT	5 (20)	1 (6.7)	4 (40)
SBRT	17 (80)	13 (87)	4 (40)
Oligometastatic disease, n (%)	8 (32)	7 (46)	1 (10)

**Table 2 cancers-16-03105-t002:** Progression-free survival and overall survival analysis in group 1, group 2, and all cases.

	Cerebral PFS (Months)	Visceral PFS (Months)	Median PFS (Months)	Median OS (Months)
Group 1 n= 15	12	20	9	NR
Group 2 n= 10	5	6	3	6
All cases n= 25	10	13	6	25

**Table 3 cancers-16-03105-t003:** First-line single or dual immunotherapy. Focus on pivotal clinical trials. NS not specified, sq: squamous, nsq: non-squamous.

	ICI arm	Patients with BM in ICI Arm (n)	Histology	BM Status	PD-L1 (%)	Median PFS (Months)	Median OS (Months)	Type of Study
Kilickap et al., 2023 [17] EMPOWER-Lung 1 BM analysis	Cemiplimab	34	NSCLC	Asymptomatic/pretreated	>50	12.5	NR	Phase III prospective
Lee et al., 2023 [18] IPSOS study	Atezolizumab	27	NSCLC	Asymptomatic/pretreated	Any	NS	HR = 0.85 95%CI [0.4,1.8]	Phase III prospective
Mansfield et al., 2021 [19]	Pembrolizumab	199	NSCLC	Asymptomatic/pretreated	1–49 or >50	4.1 in PDL1 > 50 2.3 in PDL1 > 1	19.7 in PDL1 > 50 13.4 in PDL1 > 1	Pooled analysis
Reck et al., 2016 [6] KEYNOTE 024	Pembrolizumab	18	NSCLC	Asymptomatic/pretreated	>50	HR 0.55, 95% CI [0.2–1.56]	NS	Phase III prospective
Mok et al., 2019 [20] KEYNOTE 042	Pembrolizumab	35	NSCLC	Asymptomatic/pretreated	>1	NS	NS	Phase III prospective
Carbone et al., 2017 [7] CHECKMATE026	Nivolumab	33	NSCLC	Asymptomatic/pretreated	>1	NS	NS	Phase III prospective
Reck et al., 2023 [21] Post hoc analysis CHECKMATE 227	Nivolumab + ipilimumab	68 (part 1a and 1b combined)	NSCLC	Asymptomatic/pretreated	Any	5.4	17.4	Post hoc analysis
Ready et al., 2023 [9] CHECKMATE 817	Nivolumab + Ipilimumab	44	NSCLC	Untreated asymptomatic BM	Any	2.8	12.8	Phase III prospective
Descourt et al., 2023 [2]	Pembrolizumab	176	NSCLC	asymptomatic, pre-treated or not, or if symptomatic, pretreated	>50	9.2	29.5	retrospective
Ratkovic et al., 2023 [22]	Pembrolizumab	28	NSCLC	Asymptomatic and pre treated	>50	11.5	11.5	Retrospective
Waterhouse et al., 2021 [23]	Pembrolizumab or atezolizumab	359 (42sq + 317nsq)	NSCLC	NS	Any	NS	2.9 (sq) 14.5 (nsq)	Retrospective
Wakuda et al., 2021 [15]	Pembrolizumab	23	NSCLC	Any	>50	6.5	21.6	Retrospective

**Table 4 cancers-16-03105-t004:** First-line combination ICI and chemotherapy. Focus on pivotal clinical trials. NS: not specified, sq: squamous, nsq: non-squamous.

	ICI Arm	Patients with BM in ICI Arm (n)	Histology	BM Status	PD-L1 (%)	Median PFS (Months)	Median OS (Months)	Type of Study
Paz-Ares L. et al., 2023 [8] Updated CheckMate 9La	Nivolumab + ipililumab + chemotherapy	51	NSCLC	Asymptomatic/pretreated	Any	9.7	19.3	Phase III prospective
Nadal et al., 2023 [24] ATEZO BRAIN	Atezolizumab + CHT	40	Nsq NSCLC	Untreated/asymptomatic	Any	Cerebral PFS 6.9	11.8	Phase II prospective
Paz-Ares L et al., 2018 [25] KEYNOTE 407	Pembrolizumab + CHT	20	NSCLC	Any/stable	Any	NS	NS	Phase III prospective
Gandhi et al., 2018 [26] KEYNOTE 189	Pembrolizumab + CHT	73	Nsq NSCLC	Any/stable	Any	HR = 0.42 95%CI [0.26, 0.68]	HR = 0.46 95%CI [0.2–0.62]	Phase III prospective
Swart E. et al., 2023 [27]	ICI (+/− CHT)	30	NSCLC	Any	Any	6.6	15.7	Retrospective
Wang M et al., 2023 [28]	Pembrolizumab/nivolumab/camrelizumab/sintilimab + CHT	14	NSCLC	Any	Any	6.5	15.6	Retrospective
Renaud E. et al., 2023 [29]	Pembrolizumab + CHT	29	Nsq NSCLC	Asymptomatic and pretreated	Any	10.9	NS	Retrospective
Waterhouse et al., 2021 [23]	ICI + CHT	510 (42 sq + 468 nsq)	NSCLC	NS	Any	NS	6.7 sq 10.8 nsq	Retrospective
Sugawara et al., 2021 [30]	Nivolumab + Bevacizumab + CHT	36	Nsq NSCLC	Pretreated	Any	10.5	NS	Phase III prospective

## Data Availability

The original contributions presented in the study are included in the article, further inquiries can be directed to the corresponding author.

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
