# Peer review of "Brain Metastases as Inaugural Sign of Non-Small Cell Lung Carcinoma: Case Series and Review of Literature"

_cancers, 2024, doi:10.3390/cancers16173105_

Round 1
Reviewer 1 Report
Comments and Suggestions for Authors
The authors present a well written case series and literature review of brain metastasis as presenting sign in NSCLC with treatment implications.
The case series portion of the manuscript is concise and well thought, though there is a significant limitation in conclusions due to the small sample size.
Would the authors be able to share what treatments did the patients in group one on progression receive, given the marked difference in survival amongst the two groups? Could the authors include that into their conclusions?
The literature review portion of the manuscript is excellent.
Author Response
Comments 1/2:
The authors present a well written case series and literature review of brain metastasis as presenting sign in NSCLC with treatment implications.
The case series portion of the manuscript is concise and well thought, though there is a significant limitation in conclusions due to the small sample size.
Response: Thank you for your positive comments on our work. We recognize the limitation posed by the small sample size and agree that it restricts the strength of our conclusions. We plan to pursue further research with larger cohorts to build on these findings.
Comment 3: Would the authors be able to share what treatments did the patients in group one on progression receive, given the marked difference in survival amongst the two groups? Could the authors include that into their conclusions?
Response: We have added the requested information regarding post-progression treatments in the "Patient Population" section, page 2, last 5 lines: "Post-progression treatments in Group 1 included subsequent radiotherapy (WBRT, n=2; SBRT, n=2), systemic therapy with single-agent chemotherapy (n=5), either alone or in combination with bevacizumab (n=2), platinum-based combination chemotherapy (n=1), and the KRAS inhibitor sotorasib (n=2)."
Additionally, we acknowledge that another important question to consider is whether the difference in survival between the two groups could be attributed, at least in part, to the different treatments patients received after progression, such as subsequent radiotherapy or targeted therapy. A more extensive patient population is necessary to thoroughly explore this hypothesis. We have incorporated this observation into the Discussion section, paragraph 4.
Reviewer 2 Report
Comments and Suggestions for Authors
The authors investigate the role of immunotherapy in NSCLC with brain metastases. The choice of topic for the article is very timely; modern therapies have significantly improved the life prospects of patients with lung cancer and brain metastases. The authors aim to provide an overview of the current clinical trials, highlighting the value of immunotherapy.
The authors collected data in one center. The study's limitation is the small number of relatively heterogeneous patient groups.
This paper provides an overview of opportunities in systemic therapy, both immunochemo- and immunotherapy. Its presentation follows a logical timeline from standard treatment to the future, promising possibilities even if they are under study status.
Overall, the article was well-written and organized. The references are appropriate, especially the newest ones. The strengths of the manuscript are that despite the limitations, all the closed studies have been evaluated, and the preliminary results of ongoing studies give helpful information for treating non-small cell lung cancer with brain metastases.
Author Response
Response: Thank you for your positive feedback and for highlighting the strengths of our manuscript. We acknowledge the limitation regarding the small and heterogeneous patient groups, and we are committed to further research with larger, more homogeneous cohorts in future studies. Thank you again for your thoughtful review.